



# A 1 km Hourly High-Resolution 3D Wind Field Dataset over the Yangtze River Delta Incorporating Dynamical Downscaling, Observational Assimilation, and Land Use Updates

Zhengyan Zhang[1,2], Yan-An Liu[1,2*], Xinjian Ma[1,2], Zhenglong Li[3], Pengbo Xu[4], Juan Zhang[5], Min Min[6], Di Di[7], Bo Li[8], and Jun Li[8]

*1 Key Laboratory of Geographic Information Science (Ministry of Education), East China Normal University, Shanghai 200241, China;*

*2 School of Geographic Sciences, East China Normal University, Shanghai 200241, China;*

*3 Cooperative Institute for Meteorological Satellite Studies, University of Wisconsin-Madison, Madison, Wisconsin 53706, USA;*

*4 School of Mathematical Sciences, Key Laboratory of MEA (Ministry of Education), Shanghai Key Laboratory of PMMP, East China Normal University, Shanghai 200241, China;*

*5 Shanghai Zhangjiang Institute of Mathematics, Shanghai, China.*

*6 School of Atmospheric Sciences and Guangdong Province Key laboratory for Climate Change and Natural Disaster Studies, Sun Yat-sen University and Southern Laboratory of Ocean Science and Engineering, Zhuhai 519082, China;*

*7 Collaborative Innovation Center on Forecast and Evaluation of Meteorological Disasters, Nanjing University of Information Science and Technology, Nanjing 210044, China;*

*8 Innovation Center for FengYun Meteorological Satellite (FYSIC), National Satellite Meteorological Center (National Center for Space Weather), Beijing 100081, China.*

**Manuscript to be submitted to *Earth System Science Data***

*Corresponding author: Yan-An Liu (yaliu@geo.ecnu.edu.cn)



**ABSTRACT**
High-resolution three-dimensional (3D) wind field data are critical for a wide range of
applications, including wind energy assessment, low-altitude aviation, air quality modeling, and
extreme weather forecasting. Although ERA5 reanalysis remains widely used, its relatively coarse
spatial resolution (~31 km) limits its ability to capture local-scale atmospheric processes. To
address this, this study develops an hourly 3D dynamic wind field dataset with 1 km horizontal
resolution covering the Yangtze River Delta (YRD) region during the summer months (June–
August) from 2021 to 2023, namely YRD1km, generated through advanced dynamical
downscaling of ERA5 using a customized Weather Research and Forecasting (WRF) model
configuration. The methodology integrates multi-source observational nudging with high-
resolution land use parameterization to enhance near-surface wind accuracy and terrain-induced
flow representation, particularly in urban clusters and mountainous areas. Validation against
ground-based observations confirms the superior performance of YRD1km over ERA5 for hourly
10-m wind components, with Mean Absolute Error (MAE) reduced by approximately 22% for U
and 26% for V, Root Mean Square Error (RMSE) reduced by 18% for U and 23% for V, and Nash–
Sutcliffe Efficiency (NSE) improved by 33% and 40%, respectively. On a daily mean basis, both
MAE and RMSE are reduced to below 0.4 m/s, and NSE reaches approximately 0.88. Spatially,
YRD1km captures finer spatial wind speed gradients and localized terrain-induced circulations
that are not captured by ERA5. Temporally, consistent accuracy improvements with approximately
20% lower hourly error variability are seen when compared to ERA5. Vertically, 42.2% accuracy
gains are observed in the near-surface layer when compared with radiosonde profiles. Moreover,
in a representative convective storm case, YRD1km captures multi-level wind structures that are
closely linked to the initiation and continuous development of deep convection, highlighting its




diagnostic advantage in high-impact weather events. Overall, the YRD1km 3D wind field dataset
and its integrated methodological framework provide a robust foundation for regional
meteorological applications, including high-resolution AI-based forecasting, renewable energy
planning, and weather risk management in rapidly developing regions such as the YRD. The
YRD1km 3D wind field dataset is available at https://doi.org/10.57760/sciencedb.23752 (Zhang
et al., 2025).

**Key words:** 3D wind field dataset; dynamical downscaling; multi-source observational

nudging; high-resolution land use; Yangtze River Delta
**1.   Introduction**

Accurate characterization of three-dimensional (3D) wind fields with high spatiotemporal

resolution is fundamental to modern meteorological services, wind energy development, and the
safe operation of low-altitude economy. Although widely used ERA5 atmospheric reanalysis
datasets are capable of providing wind field variables that exhibit temporal continuity and physical
consistency, their relatively coarse spatial resolution limits the capability to resolve regional-scale
wind field features (Hu et al., 2023; Jung and Schindler, 2022),  particularly in areas with complex
terrain and intense urbanization (Molina et al., 2021).

The Yangtze River Delta (YRD), as one of the most intensely urbanized regions in China,

exhibits evident spatiotemporal heterogeneity in local wind fields due to the combined effects of
sea-land thermal contrasts, urban heat island effects, and boundary layer turbulence (Zhang et al.,
2010). This presents significant challenges for precise wind energy resource assessment, urban
ventilation capacity diagnosis, and early warning of wind storm events. To address these
challenges, spatial downscaling of coarse-resolution reanalysis datasets has become a promising



strategy for improving regional wind field reanalysis and supporting fine-scale applications (Boé
et al., 2007; Tang et al., 2016; Zhang et al., 2020).
Spatial downscaling techniques primarily include statistical downscaling and dynamical
downscaling approaches. Statistical downscaling establishes statistical relationships between
coarse-resolution meteorological variables and local observational data (Dayon et al., 2015;
Tareghian and Rasmussen, 2013), enabling the acquisition of high-resolution wind field
information at relatively low computational costs (Zamo et al., 2016). However, such methods
often overlook the physical constraints among meteorological variables. In recent years, deep
learning has been increasingly applied to enhance the accuracy of statistical downscaling of wind
fields (Dujardin and Lehning, 2022; Dupuy et al., 2023; Höhlein et al., 2020; Lian et al., 2024; Liu
et al., 2024a; Zhang and Li, 2021). Nevertheless, incorporating physical consistency into deep
learning frameworks remains a significant challenge (Sun et al., 2024). In contrast, dynamical
downscaling employs the fundamental equations governing the atmospheric dynamics to explicitly
resolve physical processes, thereby reconstructing regional weather systems at high resolutions
(Tang et al., 2016). Its effectiveness has been demonstrated in various applications (Bao et al.,
2015; Liu et al., 2024b; Xu et al., 2021). Horvath et al. (2012) applied the Weather Research and
Forecasting (WRF) model with sub-kilometer grid spacing over mountainous regions of Nevada
and showed that dynamical downscaling significantly improved the representation of near-surface
wind speed and variability compared to coarser reanalysis products. Notably, when combined with
nudging techniques, the model's responsiveness to the actual atmospheric state is further enhanced
(Harkey and Holloway, 2013; Lo et al., 2008).
Nudging, also known as Newtonian relaxation, is a data assimilation method that introduces
forcing terms into numerical model equations to incrementally adjust model variables toward



observations or analysis fields (Hoke and Anthes, 1976). Compared with variational assimilation
methods, nudging does not require the construction of an adjoint model or the estimation of
background error covariance matrices. As a result, it offers a simpler implementation and lower
computational cost (Daescu and Langland, 2013; Lei and Hacker, 2015). Research has
demonstrated that this method has been successfully applied in the construction of several high-
resolution reanalysis datasets. For example, the MERIDA HRES (4 km resolution, hourly) (Viterbo
et al., 2024) and the BAYWRF (1.5 km resolution, daily) (Collier and Mölg, 2020) datasets both
employ the WRF model to perform dynamical downscaling on ERA5 reanalysis data. By
integrating nudging techniques, these datasets have reconstructed local wind field characteristics
for Italy and the Bavarian region of Germany, respectively. Although dynamical downscaling
demands substantial computational resources, advancements in regional model structures and
high-performance computing technologies are expected to greatly improve its feasibility for
regional complex terrain studies and non-climate research applications (Gutowski et al., 2020;
Yuan et al., 2024).

Furthermore, accurate representation of land surface parameters is another critical factor

influencing the performance of wind field dynamical downscaling. In recent years, high-resolution
land use data have been increasingly incorporated into wind field modeling to optimize surface
parameterization (De Bode et al., 2023; Fu et al., 2020; Santos-Alamillos et al., 2015). The updated
land use datasets enable more precise characterization of various land surface features such as
urban areas, mountainous regions, and water bodies, which improve simulation of terrain-induced
flows and boundary layer processes, particularly in complex terrain regions (Golzio et al., 2021;
Siewert and Kroszczynski, 2023).



In summary, this study presents the development of a 1-km hourly 3D dynamic wind field

dataset over the YRD region (YRD1km), covering the period of the summer months (June to

August) from 2021 to 2023. The YRD1km dataset is generated by applying a state-of-the-art

dynamical downscaling technique to the ERA5 reanalysis data, integrating multi-source

observational nudging, and updating land surface information with high-resolution ESA

WorldCover 2020 (EWC2020) land use data. The resulting dataset provides enhanced accuracy in

simulating near-surface winds and tropospheric dynamic structures, particularly in urban and

mountainous areas where wind variability is often high.

This study evaluates the performance of YRD1km relative to ERA5, with a focus on both

horizontal and vertical wind field accuracy. It also assesses the effectiveness of an integrated

methodology that combines dynamical downscaling, observational nudging, and updated land use

data in improving wind field simulations over regions with complex land surface characteristics

and atmospheric variability. The findings highlight the potential of YRD1km to support a wide

range of applications, such as localized weather forecasting, renewable energy planning, air quality

modeling, and urban environmental management in rapidly urbanizing areas.

**2.  Data**

**2.1 ERA5 Reanalysis Data**

The ERA5 reanalysis dataset (Hersbach et al., 2020), developed by the European Centre for

Medium-Range Weather Forecasts (ECMWF), integrates global multi-source observations

through 4D-Var data assimilation(https://doi.org/10.24381/cds.bd0915c6). It provides three-

dimensional hourly atmospheric variables (e.g., temperature, humidity, wind fields, and pressure)

with a horizontal resolution of 0.25°×0.25° (about 31km), serving as a widely adopted benchmark

in meteorological research. In this study, ERA5 supplies initial and boundary conditions for the



WRF model dynamical downscaling. Additionally, ERA5 serves as a baseline dataset for
comparative validation of YRD1km performance enhancements.
**2.2 Surface and Upper Air Weather Observations**
This study assimilates two observational datasets: (1) the NCEP ADP Global Upper Air and
Surface Weather Observations (https://doi.org/10.5065/Z83F-N512), comprising global terrestrial
stations, ocean buoys, ships, radiosondes, aircraft reports, and ASCAT satellite-derived winds from
the Global Telecommunication System (GTS), and (2) hourly data from Automatic Weather
Stations (AWS) operated by the China Meteorological Administration (CMA) (http://data.cma.cn/).
The spatial distributions of the two observational datasets over the YRD are illustrated in Figure
1a. The NCEP ADP dataset provides three-dimensional conventional meteorological
measurements from multiple observational platforms. As a complement to the NCEP ADP dataset,
the CMA AWS network delivers high-density surface observations across China, with a total of
2,169 stations—approximately six times the number of surface stations available from the NCEP
ADP dataset within the Chinese domain. This higher station density significantly enhances the
spatial representativeness of near-surface meteorological conditions in the YRD region. Using
Observation Nudging assimilation techniques, these datasets collectively correct systemic biases
in ERA5's near-surface fields within the WRF framework, enhancing the model's capacity to
resolve localized circulation patterns. The AWS data further act as a cross-validation source to
quantify YRD1km's accuracy improvements.
**2.3 High-resolution Land Cover Geographical Data**
Conventional land use datasets in WRF (USGS 1992-1993 or MODIS 2001) (Anderson et
al., 1976) are limited in their ability to reflect the rapid urban expansion and evolving land surface
characteristics of the YRD region. To address this, we integrate the EWC2020 dataset—a global



land cover product with 10-meter spatial resolution that classifies 11 surface types (e.g., built-up
areas, croplands, water bodies) ( https://esa-worldcover.org/en). By updating WRF's land surface
parameters with EWC2020, we refine the representation of aerodynamic roughness lengths and
urban heat island effects. For instance, reclassifying Shanghai's Pudong district from Moderate
Resolution Imaging Spectroradiometer (MODIS) "mixed urban" to EWC2020 "high-intensity
built-up" improves wind field simulations by better capturing drag effects from high-rise structures,
as validated against AWS observations.
**3. Methods**
**3.1 WRF Model Configuration for Dynamical Downscaling**

This study employs the WRF-ARW model (v4.4.2) (Skamarock et al., 2019) to establish a

dynamic downscaling framework, enhancing the spatial resolution of ERA5 reanalysis data from
~31 km to 1 km. The model domain is configured with a triple-nested grid centered at (29.36°N,
115.65°E) with horizontal resolutions of 9 km (D01, with 342×305 grid points), 3 km (D02, with
529×640 grid points), and 1 km (D03, with 919×949 grid points). The innermost domain, D03,
covers the entire YRD region (Figure 1a) and is designed to capture local circulation features
associated with urban clusters, lakes, and hilly terrain at a kilometer-scale resolution. In the vertical
direction, 61 terrain-following eta levels are used, with the model top set at 10 hPa, which
facilitates a detailed resolution of boundary layer dynamics. Through sensitivity testing, the
following physical parameterization schemes were selected: the Thompson microphysics scheme
(Thompson et al., 2008), which is well-suited for high-resolution cloud microphysics; the Dudhia
shortwave radiation (Dudhia, 1989) and RRTM longwave radiation schemes (Mlawer et al., 1997)
for radiative transfer; and for boundary layer and land surface processes, the YSU non-local closure
scheme (Hong et al., 2006) coupled with the Noah land surface model (Tewari et al., 2004), which



enhances the representation of near-surface turbulent exchanges. The Kain-Fritsch cumulus
parameterization scheme (Kain, 2004) is applied only in the outer grid (D01) to mitigate
uncertainties in the "gray zone" below the 3 km grid resolution.
To reduce the accumulation of model errors, a cold-start strategy is implemented, with
simulations initiated four times daily at 00, 06, 12, and 18 UTC, respectively. Each run generates
a continuous 6-hour forecast period, from which the first hour is discarded as model spin-up.
Ultimately, this approach produces a continuous hourly three-dimensional wind field dataset.
**3.2 Conventional Observational Data Assimilation via Nudging**
While WRF dynamical downscaling enhances dataset resolution and preserves dynamical
constraints and physical consistency, it struggles to capture fine-scale wind field features over
complex underlying surfaces (e.g., urban clusters, water bodies) without dense observational
constraints. To address this, this study employs the Four-Dimensional Data Assimilation (FDDA)
technique, integrating conventional observations and ERA5 reanalysis fields through a Nudging
approach, thereby balancing localized dynamical processes and large-scale circulation consistency.
The core formulation of this approach is:
$$\frac{\partial x}{\partial t} = F(x) + G \cdot W(t) \cdot (x_{obs} - x) \tag{1}$$

where $x$ represents the model variable, $F(x)$ denotes the model dynamical equations, $G$ is the
relaxation coefficient, and $W(t)$ is the temporal weighting function.
This study adopts a hybrid Nudging scheme combining two strategies: (1) Observation
Nudging (ON): Direct assimilation of in situ observations from CMA AWS and NCEP ADP to
dynamically refine local wind fields. (2) Analysis Nudging (AN): Application of ERA5 reanalysis
fields as constraints to impose large-scale adjustments across the entire model domain hourly
(Stauffer and Seaman, 1990), preventing deviations from large-scale circulation patterns. Thus, the





combined ON+AN assimilation scheme ensures both large-scale consistency and enhanced
regional meteorological representation.

Taking the nudging experiment on June 1, 2022, as an example, the study quantitatively

evaluates wind field accuracy over the YRD against ground-based observations using three
statistical metrics: Mean Absolute Error (MAE), Root Mean Square Error (RMSE), and the Nash-
Sutcliffe Efficiency coefficient (NSE; Nash and Sutcliffe, 1970), defined as follows:
$$MAE = \frac{1}{n}\sum_{i=1}^{n}|A_i - Oi| \tag{2}$$

$$RMSE = \sqrt{\frac{1}{n}\sum_{i=1}^{n}(A_i - O_i)^2} \tag{3}$$

$$NSE = 1 - \frac{\sum_{i=1}^{n}(A_i - O_i)^2}{\sum_{i=1}^{n}(O_i - \bar{O})^2} \tag{4}$$

where $A_i$ represents simulated values from either ERA5 reanalysis or the dynamically downscaled
results, $O_i$ denotes corresponding in situ observations, $n$ is the total number of spatiotemporally
matched observation–simulation pairs. The NSE metric ranges from $-\infty$ to 1, with values
approaching 1 indicating perfect agreement between simulations and observations. As shown in
Table 1, compared to ERA5 data, the ON+AN assimilated dataset demonstrates significant
improvements across all statistical metrics for both the 10-m zonal (U10m) and meridional (V10m)
wind components. In particular, the MAE is reduced by 26% for U10m and 27% for V10m, the
RMSE is reduced by 22% for U10m and 24% for V10m, and the NSE is enhanced by 39% for
U10m and 42% for V10m. These results confirm that the ON+AN hybrid assimilation scheme
substantially enhances the precision of high-resolution wind field datasets in the YRD region.
*Table 1. Comparison of surface (10-m) wind field performance between the ON+AN experiment*

*and ERA5 reanalysis over the YRD region.*

| Variable | Sample size | MAE (m/s) | RMSE (m/s) | NSE |
| --- | --- | --- | --- | --- |
|  |  | ERA5 | ON+AN | ERA5 | ON+AN | ERA5 | ON+AN |
|---|---|---|---|---|---|---|---|
| U10m | 8107 | 1.203 | **0.894** | 1.583 | **1.239** | 0.343 | **0.597** |
| V10m | 8107 | 1.287 | **0.940** | 1.692 | **1.289** | 0.236 | **0.556** |

### 3.3 Impact of High-Resolution Land Use Data Updates

To address the impacts of rapid urbanization on wind field simulations in the YRD, this study enhances land surface characterization by updating the default MODIS 2001 land use data in the WRF model with the EWC2020 dataset at 10-meter resolution. Comparative analysis reveals substantial discrepancies between MODIS 2001 and EWC2020, particularly in Shanghai's metropolitan core (Figure 1b and 1c). The EWC2020 dataset resolves critical urban morphological features, including urban sprawl boundaries, park green spaces within city centers, and modified water-cropland interfaces, thereby more accurately capturing spatial heterogeneity in surface properties.

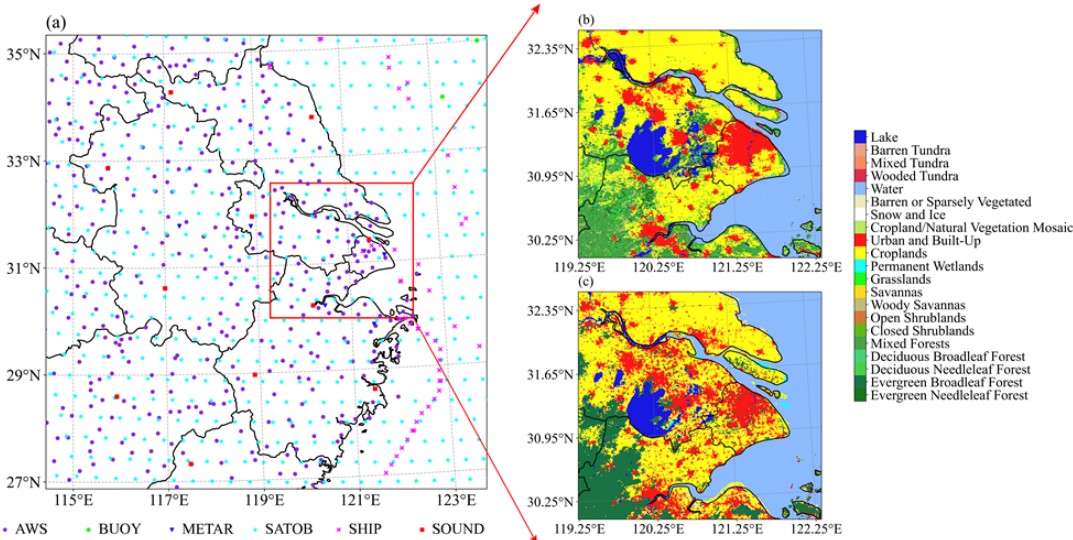

*Figure 1. Spatial distributions of key datasets used in this study. (a) Coverage of the innermost WRF domain (D03, 1-km resolution) over the YRD, along with the distribution of CMA Automatic Weather Stations (AWS) and the spatial coverage of NCEP ADP multi-source conventional*



*observations used for nudging assimilation. The red box shows the region to highlight (b) Land*
*use classification from the default MODIS 2001 dataset in WRF and (c) Updated high-resolution*
*land use classification based on the EWC2020 product.*
To quantify land use update effects on wind field simulations, we conduct two experiments
under the ON+AN assimilation framework: 1) LU-MODIS: Retains default MODIS-based land
use types; 2) LU-ESA2020: Incorporates the refined ESA2020-derived surface parameters. Using
the June 1, 2022 case study, validation metrics (Table 2) demonstrate small but obvious positive
impacts across all metrics for the LU-ESA2020 experiment compared to LU-MODIS. These
results confirm the value of high-resolution land use updates in resolving urbanization-induced
land-atmosphere interactions.
*Table 2. Statistical evaluation of land use sensitivity experiments conducted over the YRD region.*

| Variable | Sample size | MAE (m/s) | | RMSE (m/s) | | NSE | |
|---|---|---|---|---|---|---|---|
| | | LUT-MODIS | LUT-ESA2020 | LUT-MODIS | LUT-ESA2020 | LUT-MODIS | LUT-ESA2020 |
| U10m | 8107 | 0.894 | **0.886** | 1.239 | **1.232** | 0.597 | **0.602** |
| V10m | 8107 | 0.940 | **0.933** | 1.289 | **1.282** | 0.556 | **0.561** |

**3.4 High-Resolution 3D Wind Field Dataset Generation**
Building on the evaluation results in section 3.2 and 3.3, this study develops a systematic
framework for generating the YRD1km dataset over the YRD region, as shown in Figure 2. In the
preprocessing stage, observational constraints for nudging were derived from the integration and
quality control (QC) of NCEP ADP and CMA AWS datasets. Surface parameterization was refined
by replacing the default MODIS 2001 land-use data with the updated ESA 2020 dataset. For model
simulation, ERA5 reanalysis provided the initial and boundary conditions for a triple-nested WRF
configuration (9 km → 3 km → 1 km). The updated surface parameters were used to optimize the
static fields, while a suite of optimized physical schemes and a cold-start initialization strategy
were applied to suppress error accumulation. A hybrid observational nudging scheme (ON + AN)



was employed to enhance the model's consistency with observed atmospheric states, resulting in
continuous hourly 3D wind vector outputs at 1-km horizontal resolution and 61 vertical levels
during the summer months (June – August) from 2021 to 2023.
Comprehensive multi-dimensional validation was conducted using both surface station
observations and radiosonde profiles. The near-surface wind performance was evaluated through
MAE, RMSE, and NSE metrics, capturing the overall, spatial, and temporal accuracy of the dataset.
In addition, radiosonde-derived wind profiles were used to assess the vertical structure of the
reconstructed fields. A dedicated case study further demonstrates the capability of YRD1km to
resolve fine-scale dynamical features, confirming its superior performance compared to ERA5 and
highlighting the effectiveness of the integrated approach in high-resolution wind field
reconstruction.
Comprehensive multi-dimensional validation was performed using both surface station
observations and radiosonde profiles. The near-surface wind simulation performance was assessed
through MAE, RMSE, and NSE metrics, to evaluate the overall, spatial, and temporal accuracy of
the dataset. In addition, radiosonde-derived vertical wind profiles were used to examine the fidelity
of the reconstructed wind field structure in the lower and middle troposphere. Furthermore, a
typical case study highlights the capability of the YRD1km dataset to capture fine-scale dynamical
features, demonstrating clear improvements over ERA5 and underscoring the effectiveness of the
integrated approach in high-resolution wind field reconstruction.



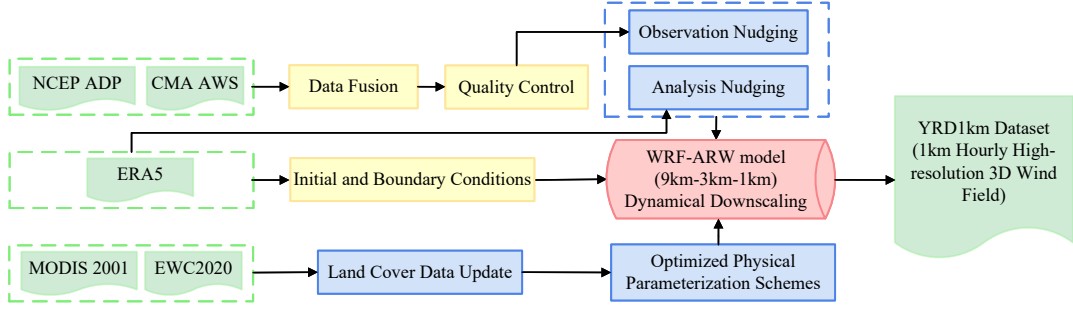


Figure 2. *The schematic workflow of YRD1km 3D wind field generation.*


**4.   Results and Discussion**
**4.1 Evaluation of YRD1km High-Resolution Dataset Accuracy**
**4.1.1 Accuracy Evaluation of YRD1km and ERA5 Based on AWS Observations**

The study conducted a comprehensive evaluation of the near-surface wind field accuracy

using YRD AWS observational data on June 1, 2022. Due to the different spatial resolutions of
YRD1km and ERA5, a nearest-grid-point matching method was adopted for comparison with
station observations (Liu et al., 2025). As shown in Figure 3, scatterplots of the 10-m wind field U
and V components for both ERA5 and YRD1km datasets were analyzed to assess their respective
simulation capabilities. Overall, YRD1km exhibited superior performance in both U and V
components, as evidenced by higher NSE coefficients, lower MAE and RMSE, and a tighter scatter
distribution. Regression slopes for YRD1km were also notably closer to the 1:1 reference line,
indicating a more accurate representation of the near-surface wind field compared to ERA5. For
the U component (Figure 3a, c), ERA5 presented an NSE of 0.34, with MAE and RMSE of 1.20
m/s and 1.58 m/s, respectively, and a regression slope of only 0.42, with increasing deviations
under higher wind speed conditions. In contrast, YRD1km achieved a significant improvement
with an NSE of 0.60, MAE reduced to 0.89 m/s, RMSE reduced to 1.23 m/s, and an increased
regression slope of 0.64, significantly reducing systematic biases. Further analysis based on the



sign of the U component revealed that ERA5 exhibited a consistent underestimation of both
easterly winds (U<0) and westerly winds (U>0), particularly under stronger wind conditions (|U|>2
m/s). This finding aligns with previous reports by Hu et al. (2023). While YRD1km also exhibited
a similar underestimation pattern, its magnitude was notably reduced, indicating an improved
representation of directional wind components compared to ERA5. Additionally, as wind speed
increased, scatter dispersion became more pronounced, with fewer samples in the high wind speed
range, adding challenges to accurate simulation.

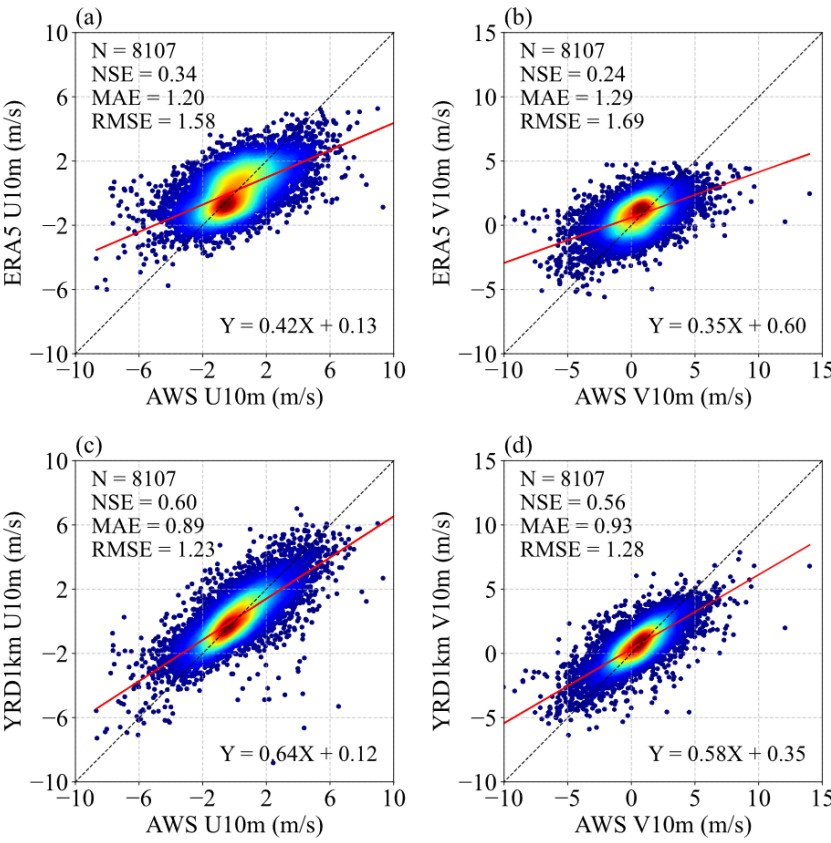


*Figure 3. Scatterplot evaluation of 10-m wind components over the YRD region: (a) ERA5*
*U10m, (b) ERA5 V10m, (c) YRD1km U10m, and (d) YRD1km V10m.*



For the V component (Figure 3b, d), ERA5 showed an even lower NSE of 0.24, with MAE
and RMSE of 1.29 m/s and 1.69 m/s, respectively, and a regression slope of 0.35, indicating a less
accurate simulation. Conversely, YRD1km significantly improved the NSE to 0.56, reduced MAE
to 0.93 m/s, RMSE to 1.28 m/s, and increased the regression slope to 0.58. Similar to the U
component, the V component displayed a directional-dependent error pattern, with an
underestimation of both northerly winds (V<0) and southerly winds (V>0), especially under
stronger wind conditions. The increasing scatter dispersion and simulation uncertainty with higher
wind speeds further highlight the challenges and needs of reproducing complex wind fields.
Results in Figure 3 are based on hourly data. Considering that climate research emphasizes
the use of daily data to smooth short-term fluctuations and reveal long-term trends (Kotlarski et
al., 2019; Nashwan et al., 2019; Zhang et al., 2024), this study further examined the simulation
accuracy of 10-m wind filed  at the daily mean scale. The comparison results based on daily mean
observations from 332 AWS stations in the YRD region (Table 3) demonstrate that YRD1km
maintains a stable accuracy advantage over ERA5 across all evaluated metrics for the U and V
components, as well as wind speed at 10-m height. Notably, the daily mean values of the U and V
components exhibited better statistical performance than their hourly counterparts, as temporal
averaging effectively mitigates short-term fluctuations and random errors, enhancing simulation
stability. Additionally, compared to 10-m wind speed (WSPD10m), the U and V components
demonstrated greater improvements in error metrics, with NSE values closer to 1. This is primarily
because wind speed is a scalar variable, while U and V components are vectors accounting for
wind direction errors. The scale-dependent improvements emphasize the application value of
YRD1km for both short-term weather monitoring and long-term climate analyses in the YRD
region.





To further assess the robustness of the YRD1km dataset, an independent validation was
performed by randomly withholding a subset of AWS station data from the nudging process.
Despite the exclusion of these stations from direct observational nudging, YRD1km still
outperforms ERA5 in terms of wind field accuracy at these independent locations (figure not
shown). This result suggests that improving the representation of small-scale surface parameters
may require a denser surface observation network to support more localized data assimilation.
*Table 3. Statistical comparison of daily 10-m wind fields between ERA5 and YRD1km datasets*
*over the YRD region.*

| Variable | Indicator | Data | | Improvement |
|---|---|---|---|---|
| | | ERA5 | YRD1km | (%) |
| U10m | MAE (m/s) | 0.543 | 0.289 | **46.67** |
| | RMSE (m/s) | 0.687 | 0.370 | **46.04** |
| | NSE | 0.608 | 0.886 | **70.09** |
| V10m | MAE (m/s) | 0.575 | 0.311 | **45.96** |
| | RMSE (m/s) | 0.750 | 0.398 | **46.84** |
| | NSE | 0.556 | 0.875 | **71.85** |
| WSPD10m | MAE (m/s) | 0.622 | 0.479 | **22.98** |
| | RMSE (m/s) | 0.814 | 0.605 | **25.70** |
| | NSE | -0.185 | 0.346 | **44.81** |

### 4.1.2 Comparison of spatial variations between YRD1km and ERA5

Building upon the preceding quantitative accuracy assessment, the study further examines
the spatial variations of near-surface wind fields represented by the YRD1km and ERA5 datasets,
as illustrated in Figure 4. Overall, while both datasets (Figure 4a and 4c) adequately capture the
large-scale spatial variations of 10-m wind speeds across the YRD, YRD1km demonstrates a
notable advantage in resolving mesoscale and local-scale wind field characteristics. Specifically,



YRD1km (Figure 4c) offers a much finer spatial representation of wind speed variations compared
to ERA5, closely aligned with observational data, particularly over complex terrain and urbanized
areas. This includes enhanced wind speed zones over large water bodies such as Lake Taihu,
realistic gradients in mountainous regions like southern Anhui and Zhejiang driven by valley flows
and orographic effects, as well as improved wind speed structures over highly urbanized areas such
as Shanghai. Furthermore, ERA5 exhibits underestimation of wind speed maxima near offshore
observation points (e.g., in the East China Sea). YRD1km mitigates these biases through
assimilation of AWS data via a nudging approach, enabling better alignment with ground truth
observations and significantly enhancing the fidelity of simulated wind fields.

These spatial advantages are further highlighted through detailed analyses of wind vector

fields. As shown in Figure 4b, ERA5 exhibits an overly smoothed wind field with limited flow
differentiation near topographic boundaries. In contrast, the YRD1km dataset presents highly
structured and terrain-conforming wind directions. Over the Shanghai metropolitan area (Figure
4d), the wind field aligns with urban morphological structures, showing clear directional deflection
near city boundaries and dense river network regions, primarily due to thermal forcing and surface
drag associated with urbanization. In the mountainous region near Hangzhou (Figure 4e), the wind
field captures pronounced curvature and flow separation that closely follow terrain contours,
effectively representing multiple terrain-induced processes such as valley and slope winds. Over
Lake Taihu (Figure 4f), YRD1km simulates a divergent wind pattern, with significantly higher
wind speeds over the lake surface relative to surrounding land, indicative of thermally driven lake–
land breeze circulations.

Collectively, the spatial patterns observed in both scalar (wind speed) and vector (wind

direction) fields strongly affirm the capability of YRD1km to resolve sub-regional atmospheric
dynamics. These results further highlight the dataset's potential for supporting a broad spectrum
of regional meteorological applications.

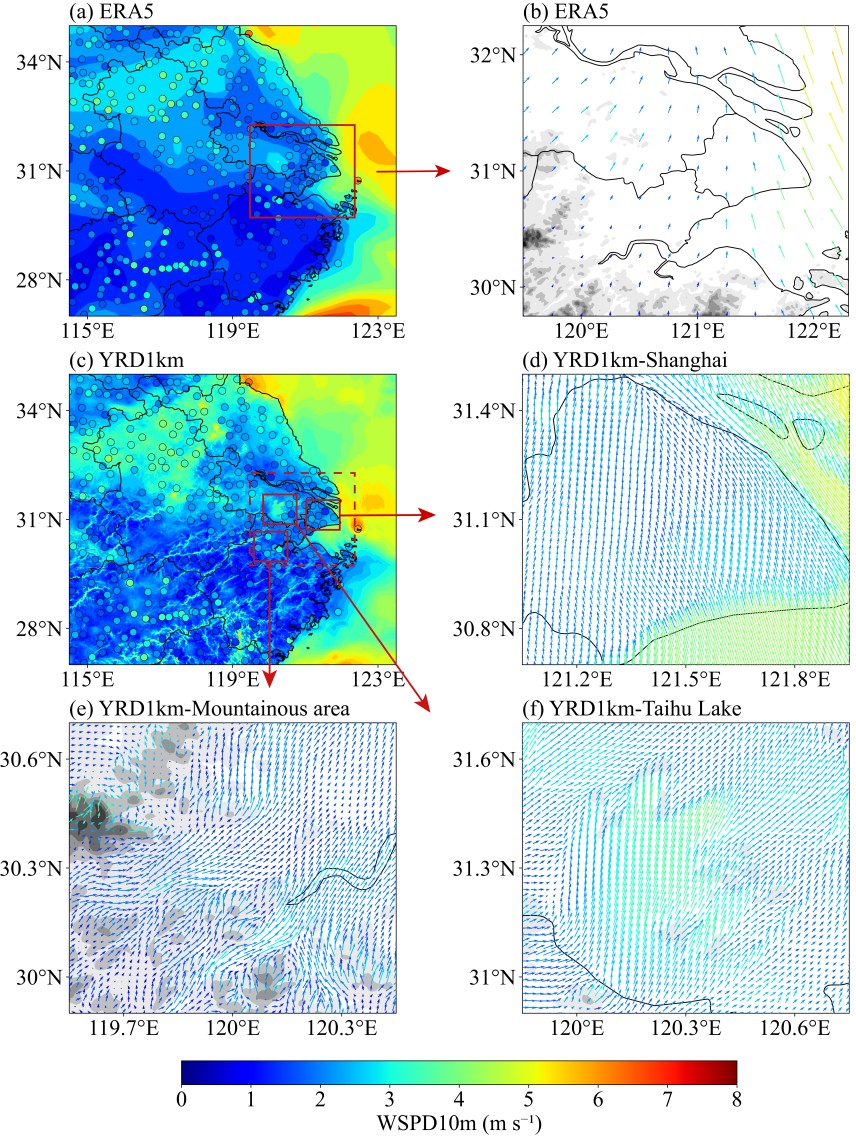


*Figure 4. Spatial distribution of daily mean near-surface wind fields over the YRD region on 1*

*June 2022. Panels (a) and (c) show daily mean 10-m wind speed (WSPD10m) from the ERA5 and*



*YRD1km datasets, respectively, overlaid with AWS station observations (colored dots). Panels (b),*
*(d), (e), and (f) show locally enlarged wind vector fields: (b) ERA5 over Shanghai and its*
*surrounding urban agglomeration; (d) YRD1km over the Shanghai metropolitan area; (e) the*
*mountainous region near Hangzhou; and (f) Lake Taihu. Arrows are color-coded by wind speed*
*magnitude and overlaid on shaded terrain elevation, with darker tones indicating higher altitudes.*
**4.2 Statistical Analysis of the Long-term Time Series of Surface Wind**
To assess the temporal performance of the proposed YRD1km dataset, hourly time series
analyses of the U10m and V10m wind components were conducted over the YRD region for June
2022. Figures 5 presents the corresponding evolutions of MAE and NSE for both wind components,
comparing the YRD1km product (red lines) with the ERA5 reanalysis (blue lines), based on
validation against ground-based observational data.
The YRD1km dataset consistently outperforms ERA5 across both components and both
metrics. MAE values for YRD1km remain consistently lower than those of ERA5, particularly
during nighttime hours, in agreement with the statistical results summarized in Table 4, which
show MAE reductions of 21.61% for U10m and 26.04% for V10m. In addition, the RMSE values
for U10m and V10m are reduced by 18.30% and 22.63%, respectively. These results indicate the
effectiveness of combining multi-source nudging and high-resolution land use data in consistently
capturing subtle wind variations over time.
Both wind components exhibit pronounced diurnal cycles in MAE, characterized by peak
errors during daytime, particularly around local noon, and reduced errors during nighttime. This
pattern reflects the influence of boundary layer dynamics, where daytime convective mixing
enhances wind variability and poses greater challenges for model accuracy, whereas nocturnal
stability leads to more predictable near-surface wind behavior. The persistence and regularity of
this fluctuation across the month highlight the necessity of capturing diurnal processes in high-
resolution simulations.

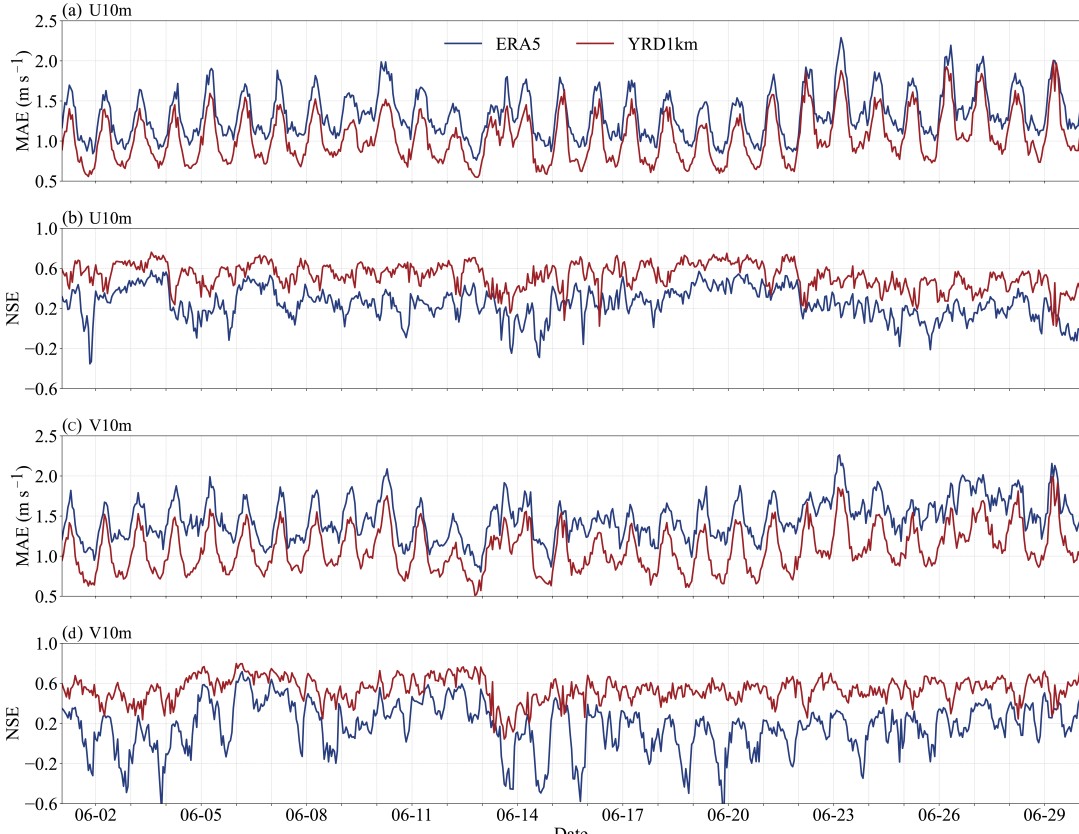


*Figure 5. Time series of model performance metrics for hourly 10-m wind components over the*
*YRD region in June 2022. Panels (a) and (b) show the MAE and NSE, respectively, for the U10m.*
*Panels (c) and (d) show the corresponding MAE and NSE metrics for the V10m. The red and blue*
*lines represent the YRD1km and ERA5 datasets, respectively.*

In terms of NSE, YRD1km maintains higher and more stable values throughout the month

for both U10m and V10m. Specifically, NSE values increase by 33.27% for U10m and 40.13% for
V10m compared to ERA5. While ERA5 frequently exhibits degraded performance, including
negative NSE values during high-variability periods, YRD1km often sustains NSE above 0.4, with



frequent peaks exceeding 0.6, especially during nocturnal hours. This reflects a markedly
improved temporal agreement between modeled and observed wind variations.

Overall, the consistent improvements observed across both horizontal wind components

confirm the robustness of the proposed downscaling framework. By effectively addressing both
synoptic-scale and diurnal-scale variability, the YRD1km dataset provides a substantially
enhanced representation of near-surface wind fields in a complex and highly urbanized region such
as the YRD.
*Table 4. Evaluation of 10-m wind field simulation performance over the YRD region in June*

*2022.*

| Variable | Sample size | Indicator | Data | | Improvement |
| --- | --- | --- | --- | --- | --- |
| | | | ERA5 | YRD1km | (%) |
| U10m (m/s) | 243280 | MAE | 1.333 | 1.045 | **21.61** |
| | | RMSE | 1.766 | 1.443 | **18.30** |
| | | NSE | 0.468 | 0.645 | **33.27** |
| V10m (m/s) | 243280 | MAE | 1.474 | 1.090 | **26.04** |
| | | RMSE | 1.938 | 1.500 | **22.63** |
| | | NSE | 0.407 | 0.645 | **40.13** |

**4.3 Evaluation of Vertical Wind Profile Accuracy Using Radiosonde Observations**

To comprehensively evaluate the vertical simulation performance of the YRD1km dataset,

radiosonde observations from the Baoshan station in Shanghai were used for the month of June
2022 at 00 and 12 UTC. A comparative analysis was conducted between YRD1km and ERA5
reanalysis data for wind speed accuracy within the 1000–100 hPa pressure range, focusing on both
Bias and RMSE metrics. The YRD1km dataset provides outputs at 32 standard vertical levels,
ranging from 1000 hPa near the surface to 10 hPa in the upper atmosphere. Key pressure levels





include: 1000, 975, 950, 925, 900, 875, 850, 825, 800, 775, 750, 700, 650, 600, 550, 500, 450, 400,
350, 300, 250, 225, 200, 175, 150, 125, 100, 70, 50, 30, 20, and 10 hPa.
As illustrated in Figure 6a, the vertical profiles of bias (dashed lines) and RMSE (solid lines)
reveal that the YRD1km dataset outperforms ERA5 across nearly all pressure levels. The
improvements are pronounced in the lower troposphere, benefiting from the dynamic constraints
of multi-source observational nudging on near-surface winds and the refined land surface flux
representation driven by high-resolution land use data. The maximum reduction in RMSE reaches
up to 1.1 m/s at 975 hPa, representing a 42.2% improvement and highlighting the substantial
enhancement in near-surface wind speed accuracy provided by YRD1km.
Time–height cross-section of wind vector differences plot (Figures 6b and 6c) further
highlights the clear performance of YRD1km. In Figure 6b, ERA5 exhibits frequent and large wind
speed differences, often exceeding ±5 m/s, along with abrupt directional shifts, particularly within
the near-surface layer. Notably, at 00 UTC on June 24, radiosonde data indicate a sharp wind speed
increase above the 950 hPa level, exceeding 19.5 m/s, which ERA5 significantly underestimates.
This result is consistent with previous studies that have identified ERA5's limitations in capturing
extreme wind events due to its coarser resolution and less-constrained boundary layer
parameterizations (Alkhalidi et al., 2025). In contrast, the YRD1km dataset exhibits a more stable
vertical wind structure, with smaller deviations from observed values. Although slight
underestimations remain during high wind episodes, the magnitude of extreme discrepancies is
considerably reduced compared to ERA5. This improvement underscores the effectiveness of the
multi-source observational nudging system in locally constraining vertical wind profiles and
enhancing model fidelity.





In summary, the YRD1km dataset, developed through the synergistic integration of high-
resolution land surface information and multi-source data assimilation techniques, significantly
improves not only near-surface wind simulations but also the representation of vertical wind
structures. This provides a reliable, high-quality data foundation for a wide range of 3D wind field–
dependent applications, such as low-level wind shear, wind turbine load estimation, pollutant
cross-layer transport modeling, and urban atmospheric environment studies.

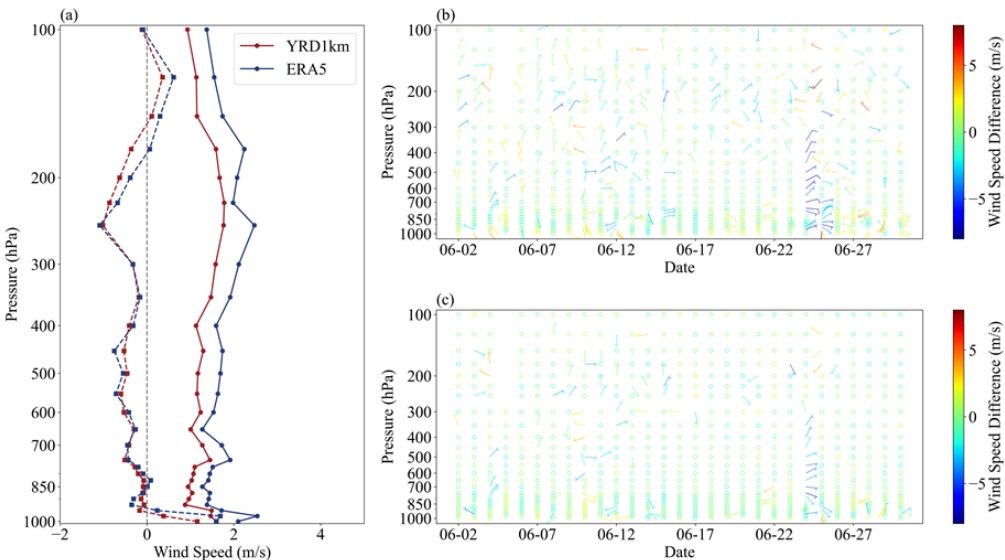


*Figure 6. Vertical evaluation of wind field performance from the YRD1km and ERA5 datasets*
*against radiosonde observations at the Baoshan station in Shanghai during June 2022. (a) Vertical*
*profiles of wind speed bias (dashed lines) and RMSE (solid lines) for YRD1km (red) and ERA5*
*(blue), calculated from all available soundings at 00 and 12 UTC. (b) Time–height cross-section*
*of wind vector differences between ERA5 and radiosonde observations (RAOB), with wind speed*
*differences (m/s) indicated by color shading. (c) As in (b), but for YRD1km minus RAOB. Wind*
*difference plots are shown at 24-hour intervals, beginning at 00 UTC on 2 June 2022.*
**4.4 Case Study of a Local Severe Convection Event**



While previous statistical validations have demonstrated the superior performance of the
YRD1km dataset spatially and temporally, its advantages become even more pronounced in short-
term, high-impact weather events. In such cases, the dataset's high spatial and temporal resolution
enhances both early warning capabilities and diagnostic accuracy.
As illustrated in Figure 7, a convective storm outbreak occurred over northern Yancheng,
Jiangsu Province, on the afternoon of 16 June 2022. The event was characterized by highly
localized and intense precipitation, with peak hourly rainfall rates reaching up to 20 mm·h⁻¹.

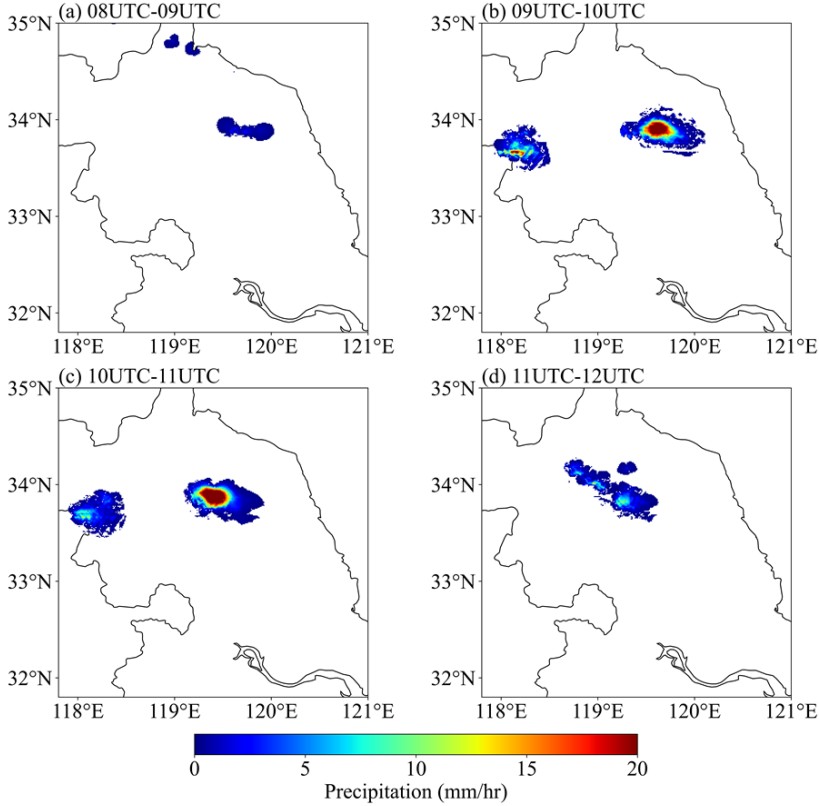


*Figure 7. Hourly evolution of precipitation associated with a convective storm over northern*
*Yancheng, Jiangsu Province, on 16 June 2022.*



To investigate the applicability of the YRD1km dataset in high-impact weather scenarios, this
study conducts a comparative analysis of wind field structures between ERA5 and YRD1km
during the convective event, focusing on three key pressure levels: 500 hPa, 700 hPa, and 850 hPa
(Figure 8). These levels are critical for identifying shear lines, low-level jets, and convective
initiation mechanisms.
Overall, the wind field structure in ERA5 appears relatively homogeneous, limiting its ability
to capture mesoscale and sub-mesoscale disturbances. In contrast, YRD1km reveals more detailed
local structures and dynamic features, demonstrating a stronger capacity to resolve mesoscale
systems. Across all three pressure levels, YRD1km consistently captures regions of enhanced wind
speed, wind shear, and convergence. Notably, near 34°N, 119°E at 500 hPa, YRD1km identifies a
localized wind speed maximum exceeding 17.5 m/s and a well-defined shear zone. At 700 hPa, a
clear convergence band and wind speed enhancement area are observed, which is conducive to the
maintenance and development of the convective system. Although wind speeds weaken at 850 hPa,
perturbation signatures remain evident. These structural features spatially align with the center of
heavy precipitation during the event, indicating that YRD1km has enhanced diagnostic capability
in capturing the dynamical background for the initiation and maintenance of deep convective
systems.
In summary, the high spatial resolution of YRD1km allows for a more accurate depiction of
wind field structures during severe convective events, thereby improving the diagnosis of key
dynamic mechanisms. This capability contributes to more effective early warning and response
strategies for short-term, high-impact weather events.







*Figure 8. Comparative analysis of wind field structures between the YRD1km and ERA5 datasets*

*during the short-duration severe convective event over Yancheng, Jiangsu Province. Displayed are*



*horizontal wind vectors (arrows) and wind speed (color shading) at the (a, b) 500 hPa, (c, d) 700*
*hPa, and (e, f) 850 hPa levels from ERA5 (left column) and YRD1km (right column) at 08:00 UTC*
*on 16 June 2022. For visual clarity, YRD1km wind vectors have been thinned by a factor of three.*
**5. Conclusions**
This study developed and rigorously validated YRD1km, a high-resolution (1 km, hourly)
3D wind field dataset over the YRD region. The dataset was generated through dynamical
downscaling of ERA5 reanalysis data using a customized WRF model configuration. It was further
refined by integrating multi-source observational nudging and updated land use representations to
improve surface parameterization.
Comprehensive validations using surface station and radiosonde observations confirmed that
YRD1km significantly outperforms ERA5 in both near-surface and vertical wind simulations. For
10-m wind fields, YRD1km consistently achieved smaller errors and higher skill scores across
MAE, RMSE, and NSE, at both hourly and daily scales. The dataset also better characterizes
spatial variability in wind speed, particularly over complex terrain and densely urbanized areas.
Its wind vector fields align well with underlying geographic features, and monthly statistics show
reductions in MAE and RMSE of approximately 20%, with NSE improved by more than 33%. In
the vertical dimension, YRD1km exhibited reduced RMSE across nearly all pressure levels and
produced observation-consistent vertical profiles. A representative severe convective case over
Yancheng demonstrated YRD1km's ability to resolve fine-scale dynamic signatures, including
wind shear, low-level convergence, and enhanced wind zones, supporting improved diagnosis of
convective development mechanisms.
These findings highlight the value of high-resolution datasets enhanced by dynamic
observational constraints in capturing both mesoscale and diurnal variability in complex





environments. The YRD1km product offers a robust foundation for enhancing meteorological
applications such as wind energy resource assessment, urban atmospheric modeling, and air
pollution transport analysis. Importantly, its fine-scale 3D wind structure also holds significant
potential for supporting the monitoring and analysis of low-level wind shear, which is critical for
the safe development of low-altitude airspace operations and the broader low-altitude economy in
urban regions.

In future work, this framework can be applied to generate longer-term high-resolution wind

datasets and extended to other regions characterized by complex terrain and heterogeneous land
use. Further enhancements may include incorporating satellite-based measurements and higher-
frequency ground-based remote sensing data, as well as coupling with machine learning models
to improve real-time forecasting and renewable energy optimization.

**Data availability**
The YRD1km 3D wind field dataset is available at https://doi.org/10.57760/sciencedb.23752
(Zhang et al., 2025).

**Author contributions**
ZZ: data collection and processing; writing (original draft preparation). YL: conceptualization;
supervision; writing (original draft preparation, review, and editing). XM, PX, and JZ: data
collection. ZL, MM, DD, BL, and JL: writing (review and editing).

**Competing interests**
The contact author has declared that none of the authors has any competing interests.




**Acknowledgments**
We gratefully acknowledge the ECNU Multifunctional Platform for Innovation 001 for providing
high-performance computing resources. We also thank the ECMWF for access to the ERA5
reanalysis data, the NCEP for the global upper air and surface weather observations, the CMA for
providing AWS data, and the ESA for the WorldCover 2020 land use data. Finally, we sincerely
thank the editor and anonymous reviewers for their insightful comments and constructive
suggestions, which greatly improved the quality of this manuscript.

**Financial support**
This research was supported by the Shanghai Science and Technology Program (Grant No.
25ZR1401099) and the National Natural Science Foundation of China (Grant No. U2142201).

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
