# Peer review of "A 1 km Hourly High-Resolution 3D Wind Field Dataset over the Yangtze River Delta Incorporating Dynamical Downscaling, Observational Assimilation, and Land Use Updates"

_Earth System Science Data, 2025_

## Author Comment (AC2)

The authors have developed a high-resolution 3D wind field dataset (YRD1km) over the Yangtze River Delta by running WRF driven by ERA5 reanalysis, assimilating observations, and updating land-use information. This dataset addresses a significant lack of high-resolution, 3D wind products in this important region during the summer months. However, the manuscript still has several structural and methodological issues. Therefore, I recommend that it be considered for publication only after major revision.

We sincerely thank the reviewer for the thoughtful and constructive comments. We have carefully considered each point and revised the manuscript accordingly. Detailed responses to all comments are provided below.

**Comments:**

1. The manuscript evaluates the dataset mainly using MAE / RMSE / NSE, but the verification metrics used in Figures 5 and 6 appear inconsistent with those used elsewhere. Unless there is a justified reason for deviation, please ensure that the verification metrics are consistent throughout the manuscript. If this is not possible, please provide an explanation in the methods section or the figure captions.

   **Response:** Thank you for this thoughtful comment. In our original manuscript, we aimed to maintain consistency in the statistical metrics and initially calculated and plotted MAE, RMSE, and NSE. Upon analyzing their temporal variations, we observed that MAE and RMSE exhibited highly similar trends. To present the results more clearly, we therefore retained only MAE and NSE in the figures. A similar relationship was observed for the vertical validation in Figure 5. We fully understand that this simplification may have caused confusion, and we sincerely appreciate your suggestion for greater consistency. In response, we have revised Figures 5 and 6 accordingly. Specifically, RMSE has been added to Figure 5, and Bias in Figure 6 has been replaced by MAE to align with the evaluation framework used elsewhere in the manuscript.

[Figure]

*Figure 5. Time series of model performance metrics for hourly 10-m wind components over the YRD region in June 2022. Panels (a), (b) and (c) show the MAE, RMSE and NSE, respectively, for the U10m. Panels (d), (e) and (f) show the corresponding MAE, RMSE and NSE metrics for the V10m. The red and blue lines represent the YRD1km and ERA5 datasets, respectively.*

[Figure]

*Figure 6. Vertical evaluation of wind field performance from the YRD1km and ERA5 datasets against radiosonde observations at the Baoshan station in Shanghai during June 2022. (a) Vertical profiles of wind speed MAE (dashed lines) and RMSE (solid lines) for YRD1km (red) and ERA5 (blue), calculated from all available soundings at 00 and 12 UTC. (b) Time–height cross-section of wind vector differences between ERA5 and radiosonde observations (RAOB), with wind speed differences (m/s) indicated by color shading. (c) As in (b), but for YRD1km minus RAOB. Wind difference plots are shown at 24-hour intervals, beginning at 00 UTC on 2 June 2022.*

2. The text in lines 242–257 appears to be duplicated. Please check the manuscript carefully and remove any redundant content.

   **Response:** We sincerely appreciate your careful reading and for pointing out this duplication. We have removed the redundant paragraph, which provides a clearer and more concise description of the validation process.

3. The manuscript's validation focuses on 10-m near-surface winds, a single radiosonde station, and an individual case study, which is insufficient to demonstrate the three-dimensional characteristics of the dataset.

   **Response:** Thank you for this valuable comment. We fully agree that a convincing

demonstration of the three-dimensional characteristics of the YRD1km dataset is essential. In the revised manuscript, we have clarified and substantially strengthened the evidence supporting its 3D performance through complementary vertical statistics, multi-station evaluations, and physically consistent case-based analyses.

As shown in the updated Figure 6a, the manuscript now presents vertical evaluations of wind fields at multiple pressure levels, revealing systematic reductions in MAE and RMSE throughout the tropospheric column (1000–100 hPa). These improvements indicate that YRD1km effectively captures both near-surface and upper-level wind structures. Together with the time–height cross-sections in Figures 6b and 6c, these results demonstrate that YRD1km reproduces the vertical stratification and temporal evolution of winds more realistically than ERA5. To further assess the representativeness of the vertical performance, we conducted similar validations from all valid sounding samples at 11 radiosonde stations across the Yangtze River Delta are now presented in Figure S1. These statistics show robust and systematic improvements of YRD1km relative to ERA5, including average reductions of approximately 30% in MAE, 28% in RMSE, and increases of about 48% in NSE, demonstrating enhanced vertical wind representation across the entire troposphere.

[Figure]

Figure S1. Vertical evaluation of wind field performance from the YRD1km and ERA5 datasets against radiosonde observations at all 11 stations across the Yangtze River Delta in June 2022. Panels show (a) MAE, (b) RMSE and (c) NSE, computed using all available 00 UTC and 12 UTC soundings.

Regarding the concern that the manuscript relies on an individual case study, we emphasize that the convective event analyzed is not intended as standalone evidence, but rather as a physically interpretable illustration of the dataset's three-dimensional dynamical consistency. To further address this concern, we additionally include an independent mesoscale convective precipitation event that occurred over the study region from 04 to 12 UTC on 10 June 2022. During the early stage (04–06 UTC), precipitation was characterized by localized and discrete convective cells, which intensified and organized into a southwest–northeast-oriented rainband during 07–08 UTC, followed by gradual weakening thereafter (08–12 UTC; Figure S2).

[Figure]

*Figure S2. Hourly precipitation evolution during the convective event on 10 June 2022 (04–12 UTC).*

Multi-level dynamical analyses reveal that YRD1km captures the evolving 3D wind structures associated with this event more realistically than ERA5. At 500 hPa, ERA5 depicts a relatively uniform westerly–northwesterly flow, whereas YRD1km resolves finer-scale mesoscale features, including enhanced wind speed gradients and weak shear, whose spatial configuration aligns well with the subsequent orientation of the precipitation band. At 700 hPa, ERA5 exhibits generally weak winds and indistinct dynamical lifting signals, while YRD1km identifies localized wind speed enhancements and weak convergence zones that spatially coincide with regions of intense rainfall. At 850 hPa, although ERA5 represents the background southeasterly inflow, the low-level jet structure and convergence patterns are

poorly defined. In contrast, YRD1km clearly resolves a low-level jet and a pronounced deceleration zone along its leading edge, forming a low-level convergence region that closely corresponds to the location of the strongest precipitation during 07–08 UTC (Figure S3). These features highlight the role of low-level jet dynamics and associated convergence lifting in the initiation and intensification of the convective system.

Taken together, the expanded multi-station vertical statistics, regionally aggregated sounding evaluations, and the physically consistent multi-level analysis of a representative convective event collectively demonstrate that YRD1km provides dynamically coherent and regionally representative three-dimensional wind fields across spatial, temporal, and vertical dimensions.

[Figure]

*Figure S3. Comparative analysis of horizontal wind field structures between the ERA5 and YRD1km datasets during the mesoscale convective precipitation event over the study region. Shown are horizontal wind vectors (arrows) and wind speed (color shading) at the (a, b) 500 hPa, (c, d) 700 hPa, and (e, f) 850 hPa levels from ERA5 (left column) and YRD1km (right column) at 04:00 UTC on 10 June 2022, corresponding to the mature stage of the convective system. For visual clarity, wind vectors in YRD1km are thinned by a factor of six.*

4. Please ensure that abbreviations are standardised throughout the manuscript (e.g., "EWC2020" and "ESA2020").

   **Response:** Thank you for your suggestion. We have carefully reviewed the entire manuscript and standardised all abbreviations related to the land use dataset. Specifically, we now uniformly use EWC2020 to denote the ESA WorldCover 2020 land use data and this form is consistently applied throughout the manuscript, including figures, tables, and text.

5. The manuscript states that YRD1km uses 61 vertical model levels, whereas line 400 refers to 32 standard levels. Are the 32 standard levels a subset of the 61 model levels, or are they obtained by interpolation? A clear description of the vertical level and the interpolation method should be provided.

   **Response:** Thank you for this helpful comment. The 32 standard pressure levels are obtained through vertical interpolation from the original 61 terrain-following eta levels of the WRF model, rather than being a direct subset. We have added a clear description of this process in Section 4.3 to clarify the relationship between the model levels and the interpolated standard levels.

6. Table 1 and Figure 3 only present results for a single day, which may be affected by temporary weather conditions. To obtain more robust conclusions, longer time periods should be included.

   **Response:** Thank you for this valuable comment. Table 1 and Figure 3 were designed as representative examples to illustrate the dataset's capability in resolving mesoscale wind

structures and near-surface dynamics on a typical summer day. We acknowledge that wind field performance may vary under different weather conditions and at different times. To address this concern and ensure the robustness of our conclusions, we have complemented the single-day analysis with long-term statistical evaluations. Specifically, Section 4.2 presents month-long hourly time series of MAE, RMSE, and NSE for June 2022 (Figures 5), covering all days and synoptic conditions during the period. These results demonstrate consistent error characteristics and performance advantages of YRD1km over ERA5 across the entire month.

Together, the long-term evaluations confirm that the results shown in Table 1 and Figure 3 are representative rather than case-specific, and that the conclusions drawn for the dataset remain robust across varying temporal and meteorological conditions.

Thank you for this valuable comment. Table 1 and Figure 3 were designed as representative examples, based on a total of 8,107 samples, to illustrate the dataset's capability in resolving mesoscale wind structures and near-surface dynamics on a typical summer day. We acknowledge that wind field performance may vary under different weather conditions and at different times. To address this concern and ensure the robustness of our conclusions, we have complemented the single-day analysis with long-term statistical evaluations. Specifically, Section 4.2 presents month-long hourly time series of MAE, RMSE, and NSE for June 2022 (Figure 5), encompassing all days and a wide range of synoptic conditions during the period. These results demonstrate consistent error characteristics and sustained performance advantages of YRD1km over ERA5 throughout the month.

Taken together, the long-term evaluations confirm that the results shown in Table 1 and Figure 3 are representative rather than case-specific, and that the conclusions drawn for the dataset remain robust across varying temporal and meteorological conditions.

7. Please specify the number of training and testing samples in Table 3 or in the methods section, and describe the sampling strategy. The results of independent validation with withheld stations should also be presented.

**Response:** Thank you for these valuable suggestions. In response, we have revised the manuscript to explicitly describe the independent validation strategy and the associated

sampling procedure.

Specifically, approximately 10% of the AWS stations were randomly withheld from the observational nudging process and reserved exclusively for independent validation. Among the total 362 AWS stations, 36 stations were withheld for validation, while the remaining 326 stations were retained to provide observational constraints for the WRF simulations. To ensure that the evaluation captures a representative range of synoptic and mesoscale meteorological conditions, the independent validation was conducted over a continuous four-day period (1–4 June 2022). Results from the withheld stations show that the YRD1km dataset consistently outperforms ERA5 in simulating the 10-m wind field (Figure S4). For the U component, NSE increases by 5.08%, while MAE and RMSE decrease by 4.80% and 3.01%, respectively, relative to ERA5. For the V component, NSE increases by 10.11%, with corresponding reductions of 8.09% in MAE and 5.14% in RMSE. In addition, the fitted relationships between simulated and observed U and V components show a noticeably closer alignment with the 1:1 reference line (Figure S4), indicating improved fidelity in reproducing near-surface wind variability. These results demonstrate that YRD1km maintains superior predictive performance even at stations that were not included in the assimilation process.

Overall, these revisions explicitly clarify the sampling strategy and present robust independent validation results, thereby reinforcing the reliability of the proposed methodological framework and the robustness of the resulting dataset.

[Figure]

*Figure S4. Independent validation scatterplots of 10-m wind components over the YRD region: (a) ERA5 U10m, (b) ERA5 V10m, (c) YRD1km U10m, and (d) YRD1km V10m.*

8. For the Baoshan station used in validation, please provide the station metadata (latitude and longitude, elevation, and underlying surface/land-use).

    **Response:** Thank you for your valuable comment. We have added the Baoshan radiosonde station metadata to Section 4.3. The Baoshan station (ID:58362) is located at 31.39° N, 121.45° E, with an elevation of 3.3 m. The site is situated in a densely built-up urban area (Urban and Built-Up), consistent with the EWC2020 land-cover classification shown in Figure 1.

9. When validating in mountainous and lake regions (e.g., Figure 4), please include terrain/lake contours or DEM information in the maps to make them clearer.

    **Response:** Thank you for this constructive suggestion. Figure 4 already included terrain

elevation shading to illustrate the mountainous topography, where darker tones indicate higher altitudes. To further improve the visual distinction of lake areas, we have added steelblue shaded overlays for major lakes (e.g., Lake Taihu). The updated figure now more clearly distinguishes mountainous and lake regions, facilitating interpretation of local wind variations.

[Figure]

*Figure 4. Spatial distribution of daily mean near-surface wind fields over the YRD region on 1 June 2022. Panels (a) and (c) show daily mean 10-m wind speed (WSPD10m) from the ERA5 and YRD1km datasets, respectively, overlaid with AWS station observations (colored dots). Panels (b), (d), (e), and (f) show locally enlarged wind vector fields: (b) ERA5 over Shanghai and its surrounding urban agglomeration; (d) YRD1km over the Shanghai metropolitan area;*

*(e) the mountainous region near Hangzhou; and (f) Lake Taihu. Arrows are color-coded by wind speed magnitude and overlaid on shaded terrain elevation, with darker tones indicating higher altitudes. Major water bodies are shaded in steelblue for clearer identification.*

10. Please ensure that the numerical precision and formatting are consistent throughout the manuscript (e.g., 26% in Lines 199 and 46.67% in Table 3).

    **Response:** Thank you for pointing out this inconsistency. We have carefully checked all numerical values throughout the manuscript and standardised the precision and formatting. Percentages are now expressed consistently with two decimal places (e.g., 26.67%) unless used as approximate descriptive values (e.g., "about 26%").

11. The data download link provided in the abstract points to files that are currently unavailable for download.

    **Response:** Thank you for raising this concern. The dataset is currently hosted on ScienceDB with a DOI link already assigned. To comply with the repository's policy and ESSD data access requirements, the dataset has been made accessible to the editor and reviewers through a private review link during the peer-review process. The dataset will be formally released and publicly accessible immediately upon the paper's publication, at the same DOI link provided in the manuscript.

---

## Author Comment (AC3)

The authors have created a high-resolution, dynamically downscaled dataset over the Yangtze River Delta called YRD1km, based on the WRF model, dynamically downscaled from the ERA5 reanalysis. The dataset is then further refined using a hybrid nudging approach, combining observational nudging and analysis nudging, to produce a generally skillful 3D wind dataset. The authors verified the dataset against surface observations primarily for June 2022, as well as against radiosonde observations, and for a convective case study event, also in June 2022. In general, the manuscript is well written and presented, but the analysis, methodology, and conclusions drawn from this work (and particularly the verification) are incomplete. I do believe this work is valid and important, but I recommend major revisions and inclusion of additional detail + verification work prior to publication.

We sincerely thank the reviewer for the careful assessment of our manuscript and for recognizing the scientific value and relevance of the YRD1km dataset. We appreciate the constructive comments regarding the completeness of the analysis, methodology, and validation, particularly with respect to the verification strategy.

In response to these concerns, we have substantially revised the manuscript by adding new experiments, expanding the validation period and scope, and clarifying key methodological details. The revised version includes more comprehensive and transparent verification analyses, as well as clearer descriptions of the model configuration and evaluation framework. Below, we provide detailed, point-by-point responses to each comment and describe the corresponding revisions made to the manuscript.

**Scientific Comments:**

- Though the dataset is valid for the summer months of 2021-2023, most of the verification was conducted in June 2022, with the first few figures and tables focusing solely on June 1, 2022. Additionally, the time period for the daily verification statistics for Table 3 is unclear; also, the time series for Figure 5 wasn't clear (are those averaged MAE/NSE across all stations?). Though I don't doubt that 1-km WRF would generally outperform the 0.25-deg ERA5 for winds, the shown verification statistics should be more transparent and comprehensive; otherwise it feels like "picking and choosing".

**Response:** Thank you for this important comment regarding the representativeness, transparency, and temporal coverage of the validation strategy.

To clarify the experimental design, June 2022 was selected a priori at the beginning of the study rather than after comparing multiple periods. In particular, 1 June 2022 was chosen as a representative early-summer day to illustrate spatial wind field characteristics, and not because it exhibited especially favorable performance.

To address the concern about transparency and to avoid any impression of selective sampling, the temporal scope of the validation analyses has been substantially expanded. For the methodology and sensitivity experiments (Section 3), all statistics originally based on a single day are now evaluated over a continuous four-day period from 1 to 4 June 2022, resulting in more than 30,000 samples for Table 1 and Table 2. For Section 4.1, the validation period has been further extended to a one-week interval from 1 to 7 June 2022, yielding nearly 60,000 samples for Figure 3. The daily statistics reported in Table 3 correspond explicitly to the same 1–7 June period and are aggregated from the hourly evaluations shown in Figure 3. The time series in Figure 5 (now Figure 6) represent spatially averaged MAE, RMSE, and NSE computed over all available surface stations across the YRD region for every hour of June 2022. These clarifications have been added to the revised text, table captions, and figure captions.

As the validation period was extended, the numerical values of the statistics changed slightly; however, the overall performance differences between YRD1km and ERA5 remain highly consistent. These revisions ensure that the presented verification results are comprehensive, transparent, and representative rather than case-specific.

- The model setup is not particularly clear, e.g. the justification of the physics parameterizations based on sensitivity testing is dubious. Those schemes are the baseline "default" schemes for the WRF model; though I understand that extensive testing of physics schemes would be unfeasible and beyond the scope of this work, many other dynamical downscaling related articles have discussed this and have either used other schemes and/or have provided justification for why they chose the schemes that they did. Additionally, the nesting and initialization setups are unclear, i.e. was two-way nesting used? Did all three domains start at the same time, meaning that the ERA5 initial

conditions would be interpolated directly to the 1-km grid? Is a 1 hour spinup really adequate? Are there discontinuities/"jumps" in the data every 6 hours?

**Response:** Thank you for this constructive comment regarding the clarity of the model configuration, the justification of the physical parameterization schemes, and the nesting and initialization strategy.

Regarding the physical parameterizations, we acknowledge that several schemes adopted in this study represent commonly used baseline configurations within the WRF modeling community. For microphysics, the Thompson scheme (Thompson et al., 2008) was selected because it is widely regarded as well suited for high-resolution simulations and has demonstrated robust performance in representing cloud microphysical processes, which is consistent with the objectives of this study.

For the planetary boundary layer (PBL) scheme, we conducted comparative tests among several commonly used options, including MYJ, MYNN, ShinHong, and YSU. As shown in Figure S2, the YSU scheme exhibits a slight but consistent advantage in near-surface wind performance relative to the other schemes. Although the differences are relatively modest and therefore were not elaborated upon in the main text, this outcome supports the selection of YSU for the final configuration. In addition, following your suggestion, we have cited recent comprehensive sensitivity studies (e.g., Sahu et al., 2025), which evaluated a large number of physics scheme combinations and confirmed that the Thompson–YSU pairing performs favorably for convective and high-resolution applications. To avoid overstatement, we have revised the manuscript wording from "Through sensitivity testing" to "Based on previous studies" when describing the selection of physical parameterizations.

Concerning the model configuration, two-way nesting was employed for all three domains. All domains were initialized simultaneously, with ERA5 initial conditions interpolated directly onto each domain, including the 1-km grid. Wind is a prognostic model variable that adjusts rapidly to the model dynamical framework, in contrast to variables such as precipitation that are indirectly generated and typically may require longer spin-up periods to equilibrate. Accordingly, a 1-hour spin-up was adopted to balance model adjustment and error accumulation. The hourly evaluation results shown in Figure 6 further indicate that no evidence of artificial discontinuities or abrupt "jumps" associated with the 6-hour cycling strategy is observed.

To improve clarity and address this comment, we have revised the Methods section to explicitly describe the nesting strategy, initialization procedure, and the rationale underlying the selection of physical parameterization schemes.

[Figure]

Figure S2. Sensitivity comparison of near-surface temperature and wind simulations using different planetary boundary layer (PBL) schemes in the WRF model. Shown are statistical performance metrics for 2-m temperature (T2m), 2-m relative humidity (RH2m), and 10-m wind components (U10m and V10m), including Mean Bias (MB), Mean Absolute Error (MAE), Root Mean Square Error (RMSE), and Nash–Sutcliffe Efficiency (NSE). Results are compared among ERA5 and four

commonly used PBL schemes (MYJ, MYNN, ShinHong, and YSU). The shaded background indicates different variable groups. Overall, the YSU scheme exhibits slightly better performance for near-surface wind variables, supporting its selection in the final model configuration.

- How much of the skillful verification statistics are due to the WRF model/enhanced spatial resolution, rather than an "edit" of grid point values based on observation nudging? If WRF grid points were nudged on the hour, and verified on the hour (and on the same grid points as the nudging? unclear in the manuscript), then there would be a reduction in error metrics via that "edit".

It would have been nice to see a baseline configuration with WRF being run without any nudging at all, to actually determine how much the nudging improved skill.

Additionally, it wasn't clear how the obs nudging was fully conducted, i.e. what variables were nudged (was it just wind?). Would the other variables (e.g. 2-m temperature, precipitation) be compromised because of this nudging, i.e. is this dataset only useful for wind, and not for other variables?

**Response:** We thank you for this important and insightful comment regarding the attribution of the skill improvements, the respective roles of enhanced spatial resolution and observational nudging, and the potential implications of the nudging strategy for other variables.

As discussed in the Methods section, we quantitatively assessed the impacts of both observational nudging and high-resolution land-use updates on model performance. When WRF is applied solely as a free-running dynamical downscaling system, its higher spatial resolution can indeed yield improved accuracy relative to ERA5 during the initial forecast hours. However, as forecast lead time increases, model errors tend to accumulate, and performance can degrade relative to ERA5, which is continuously constrained by data assimilation. By contrast, the hourly observational nudging applied in this study directly constrains the model state and therefore contributes to improved verification statistics.

To address the concern that the reported skill gains might arise primarily from a local "editing" effect at nudged grid points, we introduced an explicit independent validation design (Section 4.1.2). In this experiment, approximately 10% of the AWS stations were randomly withheld from the nudging procedure and used exclusively for independent evaluation. These stations provided no

direct observational constraints to the model. Validation over a continuous four-day period (1–4 June 2022) shows that YRD1km still consistently outperforms ERA5 at these independent locations, with systematic improvements in MAE, RMSE, and NSE for both wind components (Figure 4). The closer alignment of modeled and observed winds with the 1:1 relationship further confirms that the skill gains are not confined to assimilated grid points, but extend spatially across the domain.

These findings indicate that the improved performance of YRD1km cannot be attributed solely to pointwise nudging "edits." Instead, it reflects the combined effects of enhanced spatial resolution, physically consistent dynamical downscaling, refined land-surface representation, and the hybrid observational–analysis nudging framework. Moreover, the spatial wind field patterns shown in Figure 5 and the multi-level analyses of the representative convective case in Figure 10 further demonstrate that YRD1km captures meteorologically important mesoscale and convective features more effectively than ERA5.

Regarding the nudging configuration, we clarify that observational nudging was applied not only to wind, but also to standard near-surface thermodynamic variables, including 2-m temperature, relative humidity, and surface pressure, following conventional FDDA practice. Precipitation was not nudged, as it is not a prognostic model variable and cannot be directly constrained through nudging. While the present manuscript focuses on wind fields in line with the primary objective of constructing a high-resolution 3D wind dataset, the inclusion of thermodynamic variables in the nudging process helps maintain physical consistency within the boundary layer and does not compromise other fields. A more comprehensive evaluation of temperature and humidity fields is planned for future work.

Overall, through the addition of the independent validation experiment and clarification of the nudging strategy, the revised manuscript more clearly demonstrates that the skill improvements in YRD1km arise from physically meaningful enhancements rather than from localized nudging artifacts alone.

[Figure]

Figure 4. Independent validation scatterplots of 10-m wind components over the YRD region: (a) ERA5 U10m, (b) ERA5 V10m, (c) YRD1km U10m, and (d) YRD1km V10m.

- Are there not other studies/datasets with high-resolution downscaling over the YRD, that could be compared against? Purely from an interpolation/scale standpoint, of course the YRD1km dataset would be better able to resolve the winds than ERA5, especially if ERA5 grid points were interpolated to the same observation points for verification (i.e. multiple obs stations within the same ERA5 grid box would have the same ERA5 wind interpolated to them, but would have different WRF winds from different WRF grid points).

This ties into the utility of the dataset and the claims of such utility within the article. Statements like "The scale-dependent improvements emphasize the application value of YRD1km for both short-term weather monitoring and long-term climate analyses in the YRD region" are not backed up within the article. There just isn't enough verification conducted, over long enough time periods

+ seasons, against other competitive datasets, to make such claims. In particular, YRD1km isn't useful for "short-term weather monitoring" because it is dynamically downscaled from a reanalysis dataset (i.e. not useful as a forecast or nowcast dataset, only hindcasting), nor is it long enough yet for long-term climate analyses. I would recommend the authors be more specific about what the YRD1km can be used for, e.g. as a future training dataset for AI-based downscaling (with a longer dataset), or as input into high-resolution air-quality dispersion modelling or even sub-kilometer scale downscaling (down to LES scales), for specific applications (these were stated in the conclusion, but could be made more explicit and specific in other parts of the article).

**Response:** Thank you for this important and constructive comment regarding the role of spatial interpolation in the reported skill improvements and the scope of the dataset's stated applications.

At present, there is a lack of publicly available, kilometer-scale wind field datasets over the YRD. This data gap was a primary motivation for conducting this study. We agree that, from a purely spatial interpolation perspective, a higher-resolution product will generally yield lower errors when evaluated against dense surface observations, particularly when multiple stations fall within a single coarse-resolution ERA5 grid cell. However, the improvements demonstrated by YRD1km cannot be attributed solely to spatial interpolation of ERA5 fields. Unlike single-variable interpolation, YRD1km is produced through a fully dynamical framework that jointly constrains multiple atmospheric variables under physical laws, incorporates high-resolution terrain and land-surface information, and resolves mesoscale and vertical structures that cannot be recovered through interpolation alone. This distinction is particularly evident in the three-dimensional wind field analyses presented in the manuscript. For example, the vertical wind structures associated with the representative convective cases show clear convergence bands and wind speed enhancement zones that are absent in ERA5 and cannot be reproduced through spatial interpolation of reanalysis data. These features reflect physically coherent mesoscale dynamics rather than pointwise interpolation effects.

In particular, we acknowledge that YRD1km, being dynamically downscaled from a reanalysis product, is not intended for real-time forecasting or nowcasting and should therefore not be described as a dataset for "short-term weather monitoring." We have revised the manuscript to remove or rephrase these statements and to more accurately reflect the hindcast nature and temporal

limitations of the dataset.

Following your suggestion, we have revised the manuscript to make the application scope of the YRD1km dataset more explicit, precise, and aligned with the presented validation. The revised text now emphasizes its current suitability for high-resolution diagnostic analyses of three-dimensional wind structures, event-based investigations of high-impact weather, and detailed characterization of urban and complex-terrain wind fields. In addition, YRD1km is framed as a physically consistent background dataset for applications that require fine-scale wind information, such as air-quality dispersion modeling and wind energy resource assessment. Potential uses in data-driven or AI-based downscaling studies are now clearly presented as future applications, contingent on further temporal extension of the dataset.

These revisions ensure that the claimed utility of YRD1km is fully supported by the presented validation and aligns with the dataset's design, coverage, and intended use.

**Minor Comments:**

- Should the citation/URL of the dataset be listed directly in the abstract? (not sure on this, please double check)

**Response:** Thank you for this comment. We have carefully checked the ESSD author guidelines and confirmed that including the dataset citation and DOI directly in the abstract is appropriate and consistent with ESSD formatting requirements.

- Statements like "high spatiotemporal resolution is fundamental to modern meteorological services, wind energy development, and the safe operation of low-altitude economy..." should be backed with citations, even if seemingly self-evident

**Response:** Thank you for the suggestion. Relevant citations have now been added to support the statement.

- MODIS2001 ---> did you mean MODIS2010?

**Response:** Thank you for raising this point. The land-use dataset used in the WRF configuration is indeed the default MODIS 2001 land-use classification provided within the WRF modeling system.

- Inconsistencies with abbreviations, e.g. LU-ESA2020 in the text but LUT-ESA2020 in the table

**Response:** Thank you for noting this inconsistency. The abbreviations have now been standardized as LU-EWC2020 throughout the manuscript.

- "Figures 5 presents" --> line 360

**Response:** Thank you for pointing this out. The grammatical error has been corrected to "Figure 5 presents".